# Studying the decontamination process of an irradiated beryllium reflector in a chlorine environment

Yuliya Baklanova [ID]*

Institute of Atomic Energy Branch of the National Nuclear Center of the Republic of Kazakhstan, Kurchatov, Kazakhstan

* basalai@nnc.kz

## Abstract

Beryllium, possessing unique nuclear physical properties, is currently widely used as a material for reflector and neutron moderator blocks of research nuclear reactors. It can also be applied in fusion energy as a first-wall material and neutron multiplier. However, when beryllium is irradiated, its physical-mechanical properties deteriorate due to radiation-induced microstructural damage, the generation of tritium and helium, the activation of impurities under the radiation exposure, and the absorption of fission products, which determines the need for periodic replacement of the beryllium components in nuclear installations. Moreover, due to the relatively low abundance of beryllium, a relevant problem is its purification from radioactive isotopes for potential reuse. To date, chlorination has emerged as one of the most promising methods for purifying irradiated beryllium. This study addresses the optimization of the chemical process parameters and the isolation of the beryllium component from the resulting mixture of chlorination products, including the most active radionuclides: $^3$H, $^{60}$Co, $^{108m}$Ag, and $^{137}$Cs. Laboratory-scale experiments confirmed the effectiveness of the irradiated beryllium chlorination technology for its purification. The reduction in the activity level of beryllium and its compounds was objectively monitored using gamma and beta spectrometry methods.

## Introduction

Beryllium loses its unique properties as a neutron moderator and reflector in nuclear power plants (NPP) under the conditions of strong radiation loads. During the neutron irradiation, the material undergoes swelling [1,2], resulting in embrittlement [3,4], while the impurities contained within it become activated [5–8]. The recycling beryllium waste, which, according to preliminary estimates [9], reaches several tens of tons worldwide, would allow its reuse, thereby reducing the volume of highly radioactive waste. Current developments in beryllium recycling for nuclear energy have enabled the partial separation of radionuclides through various methods, including thermal desorption for tritium removal [10,11], chemical dissolution for extracting beryllium compounds from solution [5], and sublimation for $^{60}$Co separation [12,13]. However, these methods remain insufficient in maturity and effectiveness to establish an industrial-scale technology for producing of pure beryllium fractions.

**Data availability statement:** All relevant data are within the paper.

**Funding:** This research has been funded by the Committee of Science of the Ministry of Science and Higher Education of the Republic of Kazakhstan (Grant No. BR21882185). The funders had no role in study design, data collection and analysis, decision to publish, or preparation of the manuscript.

**Competing interests:** The authors have declared that no competing interests exist.

In this study, the decontamination of irradiated beryllium was performed using a device developed as a pilot plant prototype. The applied method, "dry" chlorination, is considered as one of the most effective, according to both theoretical and experimental data [11]. This technique enables a high degree of purification of the irradiated beryllium without requiring multiple repetition of the process. Additionally, it follows a simple implementation scheme using readily available reagents.

The process of the irradiated beryllium chlorination was investigated in an installation, equipped with a reaction chamber (chlorinator), designed as a closed-loop system. This chlorinator design allowed for precise time- and temperature-controlled interaction process between chlorine and reflector elements of the JMTR research reactor (Japan) [14–16]. The result of the research determined the average rate of surface chlorine interaction with the irradiated beryllium samples cut from the reflector blocks, approximately 0.13 mg/cm$^2$/s at a temperature of 500°C. This bonding rate of free chlorine with beryllium suggests the feasibility of implementing the process in an installation with a direct-flow reaction chamber, provided that the beryllium chlorination conditions prevent the release of chlorine and its compounds into the atmosphere.

The safe handling of irradiated material, chlorine and hydrogen compounds during the operation of such an installation were assessed both through calculation [17] and experimental studies, conducted during the development of the direct-flow reaction chamber [14–17]. These studies helped determine the optimal conditions for the chemical decomposition of irradiated beryllium reflectors in a chlorine medium, ensuring both process efficiency and safety. The research also defined the key parameters of the phase separation process, ultimately release of beryllium chloride free from radioactive impurities.

## Materials and methods

The possibility of the irradiated beryllium decontamination was studied on a fragment cut from the central part of the beryllium rod of a beryllium reflector of the JMTR research materials science reactor. The sample parameters are as follows: diameter, 30 × 54 mm; weight, 70.5 g; and density, 1.84 g/cm$^3$. According to the specification [18], the initial elemental composition of the beryllium reflector is approximately 98.4% Be, 1.3% BeO, and 0.3% impurities.

The reflector was operated from 1968 to 1975, while the energy release in the reactor was 24017.4 MW day [17]. For a 50 MW JMTR reactor, the thermal neutron flux density is ∼ 8.0 × 10$^{13}$ n/(cm$^2$ s), and, accordingly, fast neutrons are ∼ 7.5 × 10$^{12}$ n/(cm$^2$ s) [9,12,13,17]. The main radioactive impurities are formed by neutron irradiation. The sample's radioactive contamination is primarily determined by radionuclides of $^3$H (2.41 × 10$^{10}$ Bq [18]), $^{60}$Co, and $^{137}$Cs (Fig 1). It should be noted that during long-term storage of irradiated beryllium, residual radioactivity is largely governed by the presence of the $^{108m}$Ag radionuclide. This is due to its significantly longer half-life (438 years) compared to other radionuclides such as $^{60}$Co (5.27 years), $^3$H (12.3 years), and $^{137}$Cs (30.17 years) [19]. Therefore, the removal of $^{108m}$Ag is also an important issue.

The irradiated reflector fragment decontamination was carried out on a specially designed installation. It is a prototype of a pilot plant [20], designed for loading and processing one reflector rod, weighing approximately 1 kg in 30 chlorination cycles, that will amount to 5.5 hours (Fig 2).

The plant main element is a reaction chamber, which is a quartz tube hermetically connected to a nickel filter, heat exchanger systems and pipelines. The beryllium reflector rod is placed inside the chamber, and the working gas is passed through it.

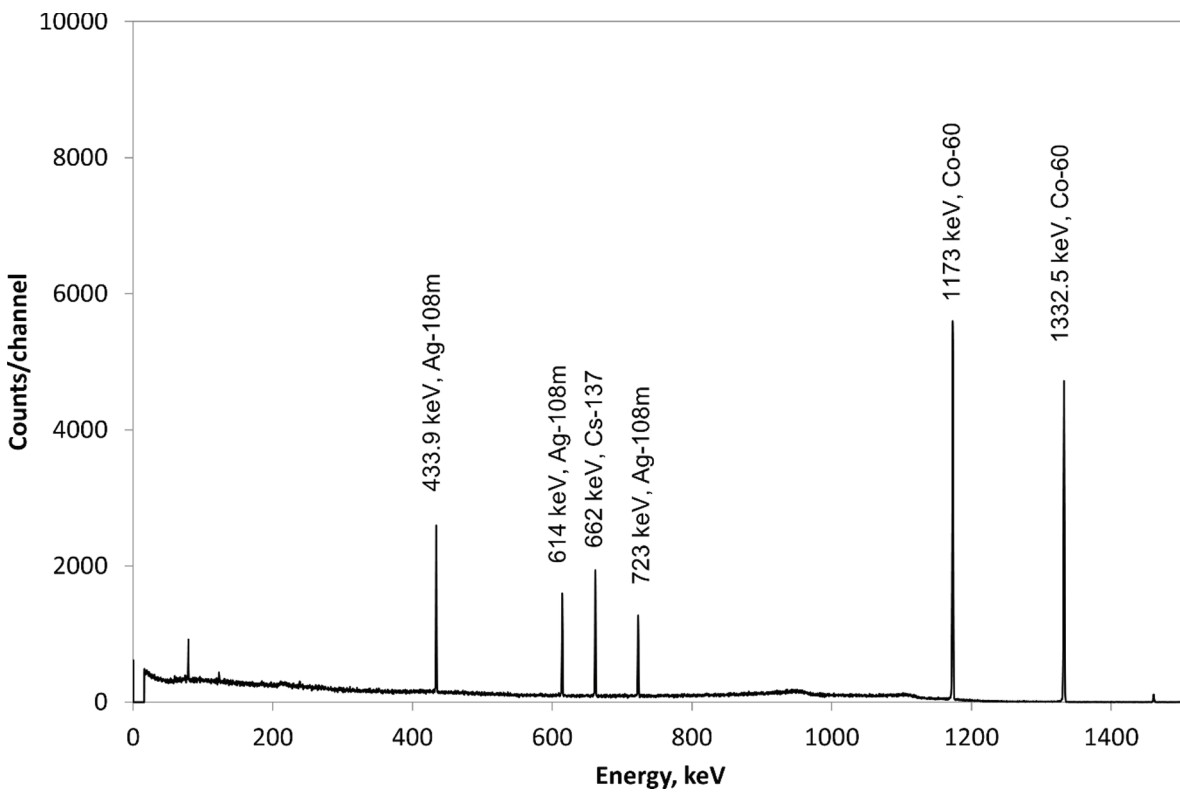

**Fig 1. Gamma radiation spectrum of the irradiated beryllium sample.**

The installation's performance is calculated based on the fact that the operation of the proposed device consists of two phases: active and passive. After heating each unit of the installation to operating temperatures is supplied into the reaction chamber during the active phase of operation. The hot beryllium chloride flow reaches the filter, gradually increasing the temperature of its working elements. The active phase ends when the temperature of the filter's working elements reaches 700°C. During the passive phase, chlorine supply is stopped, and the direct-flow system is cooled with argon, The selection of operating temperatures is based on the phase transition temperatures of the obtained chlorides [22,25]; the boiling point of beryllium chloride is 500°C, and the melting point of cobalt chloride is 724°C. The duration of the active phase was determined by the time required to heat nickel rods by 200°C in a flow of hot beryllium chloride, which was 6 minutes, while the duration of the passive phase is 5 minutes.

The beryllium reflector sample mounted on a special support assembly in the reaction chamber is preheated to a temperature of approximately 670°C, after which argon is supplied to the system. In the process of supplying argon to the installation, the temperature of beryllium is brought to the target value, 730°C, and the supply of chlorine starts (argon performs the chlorine carrier function).

The irradiated beryllium in the chlorinator is heated to a set temperature by a heater, its working elements are located on the chlorinator body outer surface. The use of a high-frequency heater provides direct heat transfer to the beryllium. The ohmic heater transfers heat through the quartz glass and the working fluid.

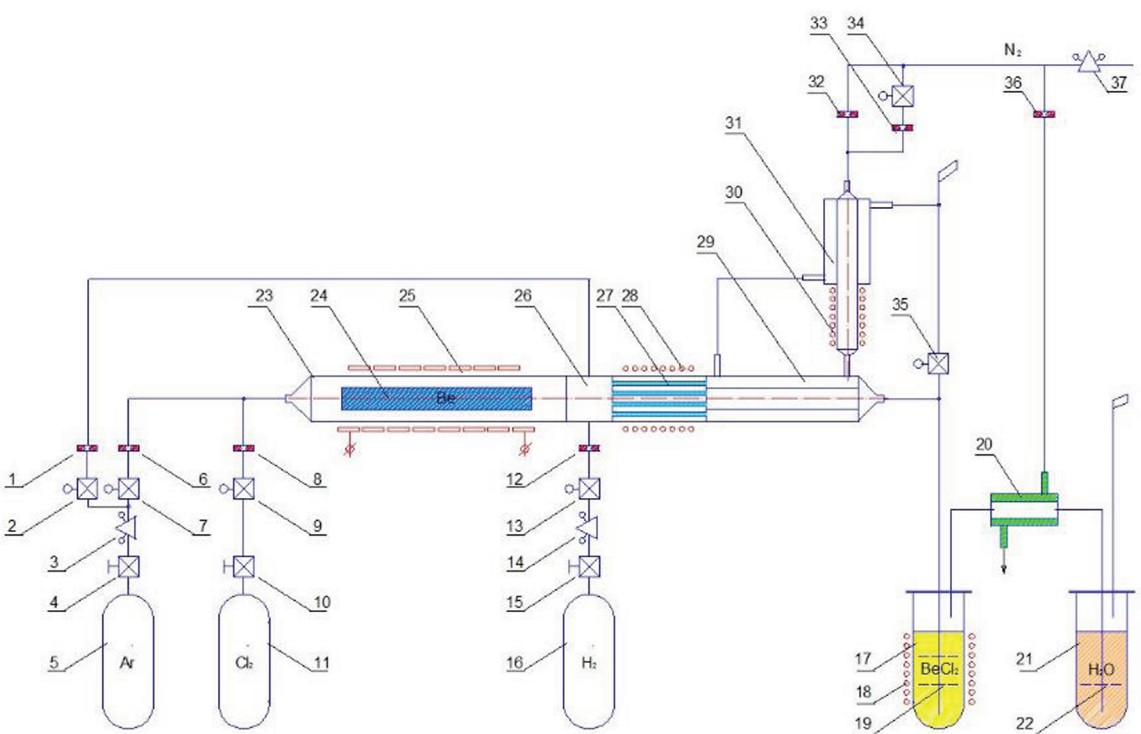

**Fig 2. Diagram of the irradiated beryllium chlorination plant.** 1, 6, 8, 12, 32, 33, 36––throttling device; 2, 7, 9, 13, 34––electric valve; 3, 14, 37––reducer; 4, 10, 15––manual valve; 5, 11, 16––gas cylinder; 17––beryllium chloride storage tank; 18, 28, 30––ohmic heater; 19, 22––gas flow distributor, 20, 29, 31––heat exchanger; 21––tritium chloride storage tank; 23––reaction chamber (chlorinator); 24––irradiated beryllium reflector; 25––high frequency or ohmic heater; 26––gas mixer; 27––nickel filter.

The radionuclide chlorides, in particular [60]Co, [137]Cs, [108m]Ag, and [3]H are formed along with the beryllium chloride under the conditions of the specified operating temperatures. The resulting gas flow is fed into the mixer. The chlorine unreacted in the reaction chamber bonds with hydrogen in the mixer forming hydrogen chloride. The hydrogen is supplied in excess to eliminate the chlorine presence in the reagent mixture completely. The temperature of the reagents in the mixer increases due to the exothermic reaction, therefore the unit has a cooling jacket that allows the gas temperature to be maintained at 600°C.

When the gas mixture passes through a filter maintained at 550°C, gamma-emitting chlorides [60]CoCl$_2$, [137]CsCl, and [108m]AgCl are deposited on the filter. Meanwhile, BeCl$_2$ (beryllium chloride) continues through the heat exchanger and condenses into the liquid phase as its temperature drops to 404°C or below [5,21].

Gaseous hydrogen chloride (HCl) and tritium chloride ([3]HCl), identical in their physical and chemical properties are additionally cooled and fed into the appropriate storage tank. The tritium chloride storage tank is filled with distilled water where the gases dissolve forming hydrochloric acid, and hydrogen is discharged through the gas spillway. The installation is equipped with an automated control system to comply with the above conditions for the cleaning process implementation. The effectiveness of the proposed method for the irradiated beryllium decontamination was evaluated by measuring the beryllium reflector fragment activity before and after the chlorination, as well as the obtained products, the samples the from nickel filter elements, beryllium chloride and tritium chloride accumulators. The plant performance was evaluated on the basis of the results of the control weighing of the

beryllium reflector sample. The samples were studied for the content of $^{60}CoCl_2$, $^{137}CsCl$, and $^{108m}AgCl$ using the InSpector-2000 gamma spectrometer (CANBERRA) with a GC1518 semiconductor detector. The $^3H$ content in the water samples from the tritium chloride storage tank was determined by the liquid scintillation method on the TRI-CARB 2900 beta spectrometer (PerkinElmer), according to the international standard [21]. In the experimental studies the active phase, which is characterized by the passage of chlorine through an irradiated beryllium sample (Fig 3a), had a duration of approximately 32 minutes. The parameters of the chlorine interaction with the beryllium were determined taking into account the chlorine mass supplied to the reaction chamber and this interaction stoichiometric nature [22,23]. The calculation was performed based on the condition that 32.5 l (103 g) of chlorine was supplied, while the maximum possible beryllium amount that could react with the chlorine was approximately 13 g. The measurement of the beryllium sample weight after taking it out from the reaction chamber showed a value of 60.53 g. The calculated amount of beryllium, that could react with the chlorine in the experiment were 4.33 g. taking into account stoichiometric the reaction parameters.

## Results

The change in the beryllium sample weight was less than expected, which was presumably due to an oxide film formation on its surface (Fig 3b), that prevented the effective interaction of the beryllium with chlorine. In the future, it is possible to remove the oxide film by passing carbon tetrachloride through a sample [15]. Considering that the time of the chlorine and beryllium interaction is 2000 s, the sample outer surface area is around 38.2 cm$^2$, and the weight decrement of the irradiated beryllium sample is 8.8 g, the surface interaction rate is approximately 0.115 mg cm$^{-2}$ s$^{-1}$. This result closely matches the average surface interaction rate of beryllium with chlorine—0.13 mg cm$^{-2}$ s$^{-1}$—observed in experiments on the cyclic chlorinator installation [23].

The beryllium chloride deposits in the storage tank are clearly visible on the inner surfaces. However, due to its strong hygroscopicity, beryllium chloride transforms into a stable crystalline hydrate, $BeCl_2 \cdot 4H_2O$ [24] (Fig 3c). Characteristic deposits are also observed on the nickel filter elements (Fig 3d).

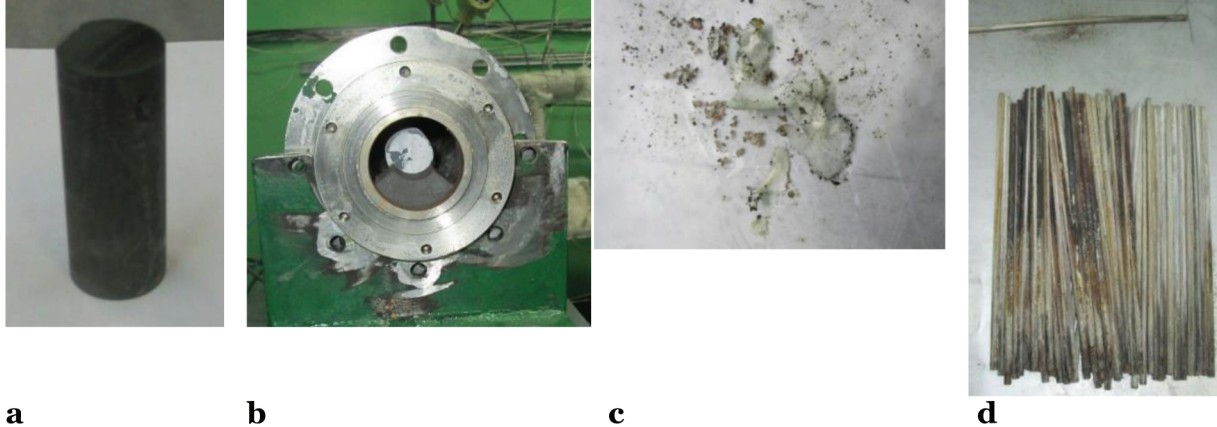

**a**　　　　　**b**　　　　　　　　**c**　　　　　　　　**d**

**Fig 3. Irradiated beryllium decontamination products.** (**a**) The reflector fragment before the chlorination; (**b**) the reflector fragment in the reaction chamber loading unit after the chlorination; (**c**) beryllium chloride sample; d – nickel filter elements after the chlorination.

Characteristic deposits are also observed on the nickel filter elements (Fig 3d). The results of the chemical analysis of sediment samples at various installation units (outlet of the reaction chamber, nickel filter, storage tanks) have shown that beryllium in $BeCl_2$ composition moves along the installation paths, which is evidence of the chemical transport reaction implementation and, therefore, indirectly indicates the implementation of the beryllium decontamination process. It has been also found that a significant amount of iron chloride is formed during the installation operation, this is due to the interaction of the material of the reaction chamber flange connections, the gas mixer, and heat exchangers made of corrosion-resistant steel 12C18N10T with the working gas. In addition, the iron chloride is transported through the gas path and deposited on the installation's internal surfaces, which makes it difficult to identify the beryllium in the surface deposits by chemical methods. The samples were taken before the filter at the reaction chamber outlet (see Fig 4), from the nickel filter elements (see Fig 5), from the beryllium chloride storage (see Fig 6), and the tritium chloride (Table 2) for a comparative analysis of the obtained results. The specific activity of $^{60}$Co, $^{108m}$Ag, and $^{137}$Cs in samples was determined from the measured intensity of gamma radiation at the total absorption peak with $E_\gamma = 1173$ keV for $^{60}$Co, $E_\gamma = 662$ keV for $^{137}$Cs, and $E_\gamma = 434$ (723) keV for $^{108m}$Ag. The total relative error in determining the specific activity of $^{60}$Co and $^{137}$Cs in samples does not exceed 6%, and it does not exceed 10% for $^{108m}$Ag with a confidence probability of 0.95.

The measurements of the gamma spectrum of the sample taken from the filter elements have shown that the activity ratio of $^{60}$Co/$^{137}$Cs increases almost five times relative to the

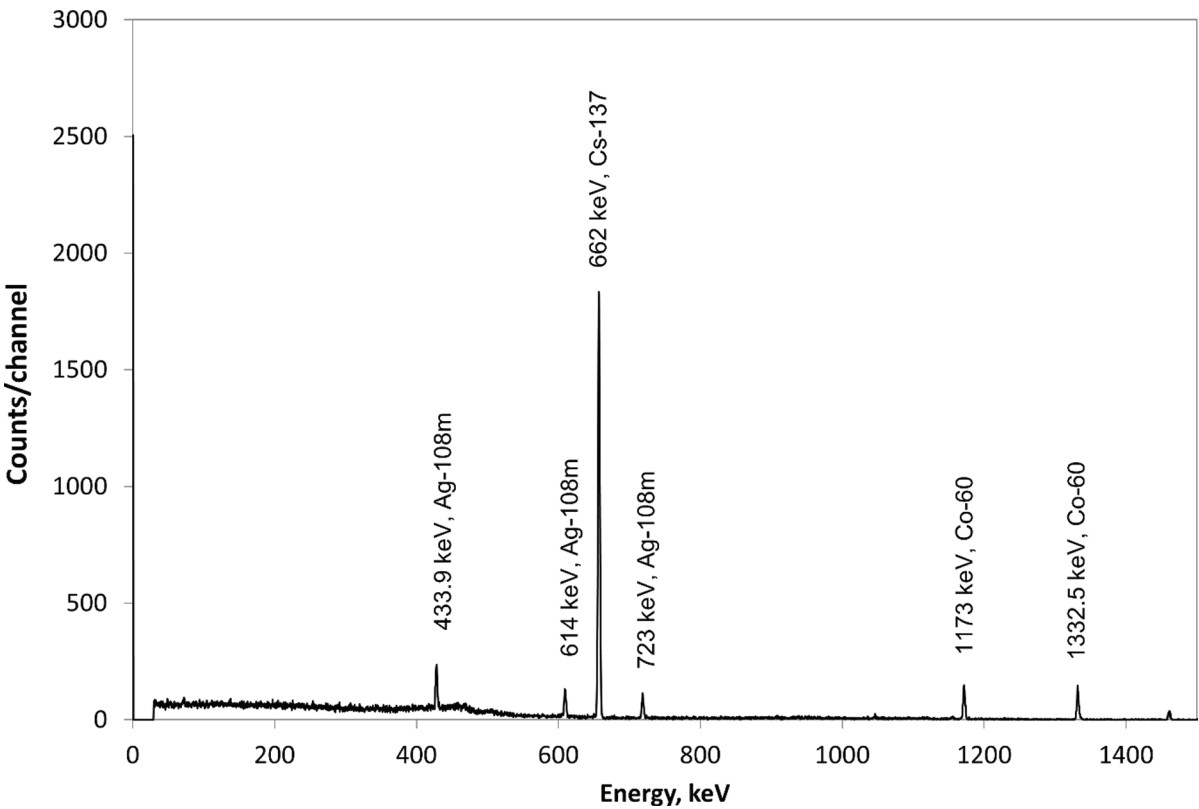

**Fig 4. Gamma radiation spectrum of the sediment sample on the reaction chamber before the filter inlet.**

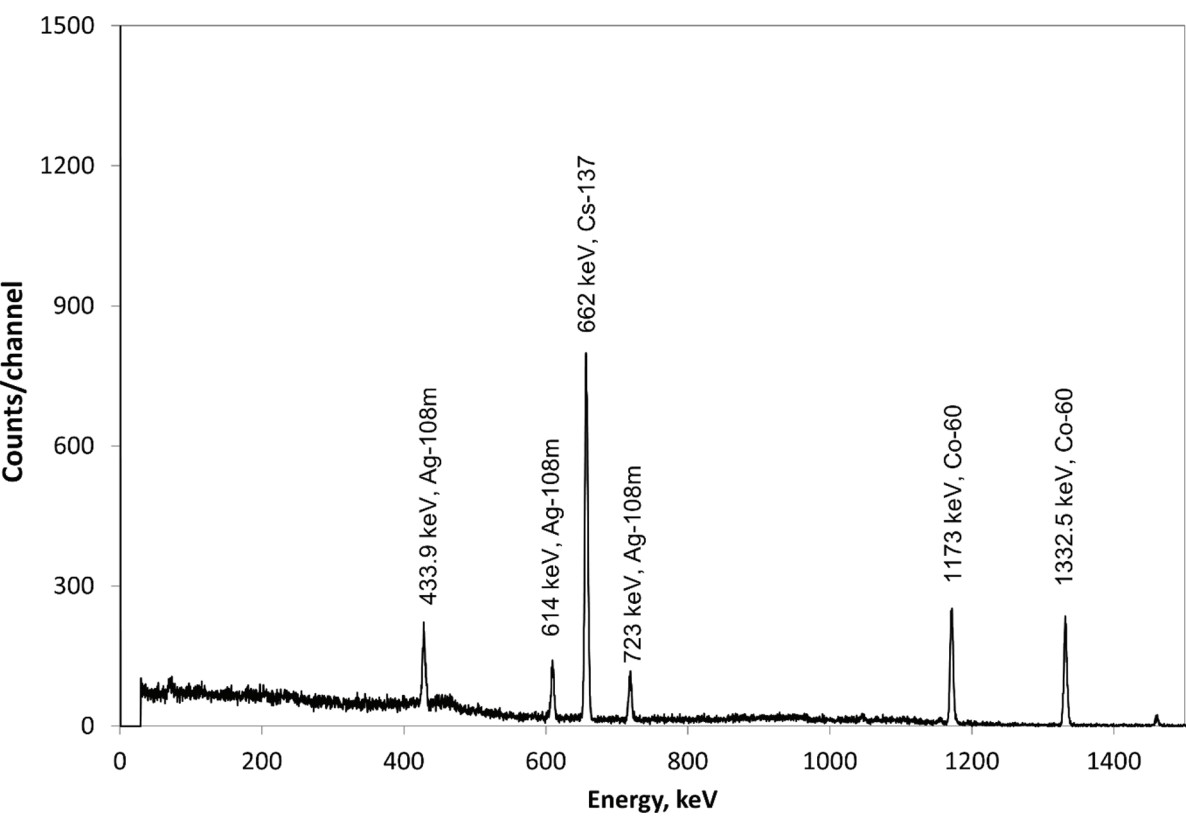

**Fig 5. Gamma radiation spectrum of the sample from the filter.**

values obtained on the samples taken before the filter inlet. This fact confirms the operability of the nickel filter as a cobalt chloride trap [14–17]. The study of the sample taken from the inner surface of the beryllium chloride storage has shown a significant decrease in the specific activity of radionuclides (Fig 6).

The results of the chemical analysis of the samples and gamma-ray spectrometry have made it possible to determine the specific activity of impurities (Bq/g(Be)) in the initial samples and the samples at the sites of the main $BeCl_2$ accumulation (Table 1).

The calculated degree of the beryllium purification from radioactive impurities was 99.6%, 99.3%, and 95.7% for the radioisotopes $^{60}$Co, $^{108m}$Ag, and $^{137}$Cs, respectively. According to theoretical estimates, the specific activity of the tritium accumulated in the beryllium irradiated in the JMTR reactor is equal to $1.7 \cdot 10^{11}$ Bq/kg [17]. Assuming that all tritium contained in the beryllium, having reacted with chlorine in the experiment, was retained in a tritium chloride storage tank in an aqueous solution (in 2 liters of water), the specific activity of the water would be $7.5 \cdot 10^8$ Bq/kg.

The latter means that 3% of the total amount of tritium contained in the irradiated beryllium, which reacted with chlorine, was retained in the tritium chloride storage.

## Discussion

The analysis of the obtained gamma-ray spectrum of beryllium chloride, as the resulting purification product of the irradiated beryllium, has shown that its residual radioactivity is determined by $^{60}$Co and $^{137}$Cs radionuclides, while the activity of these radionuclides remains

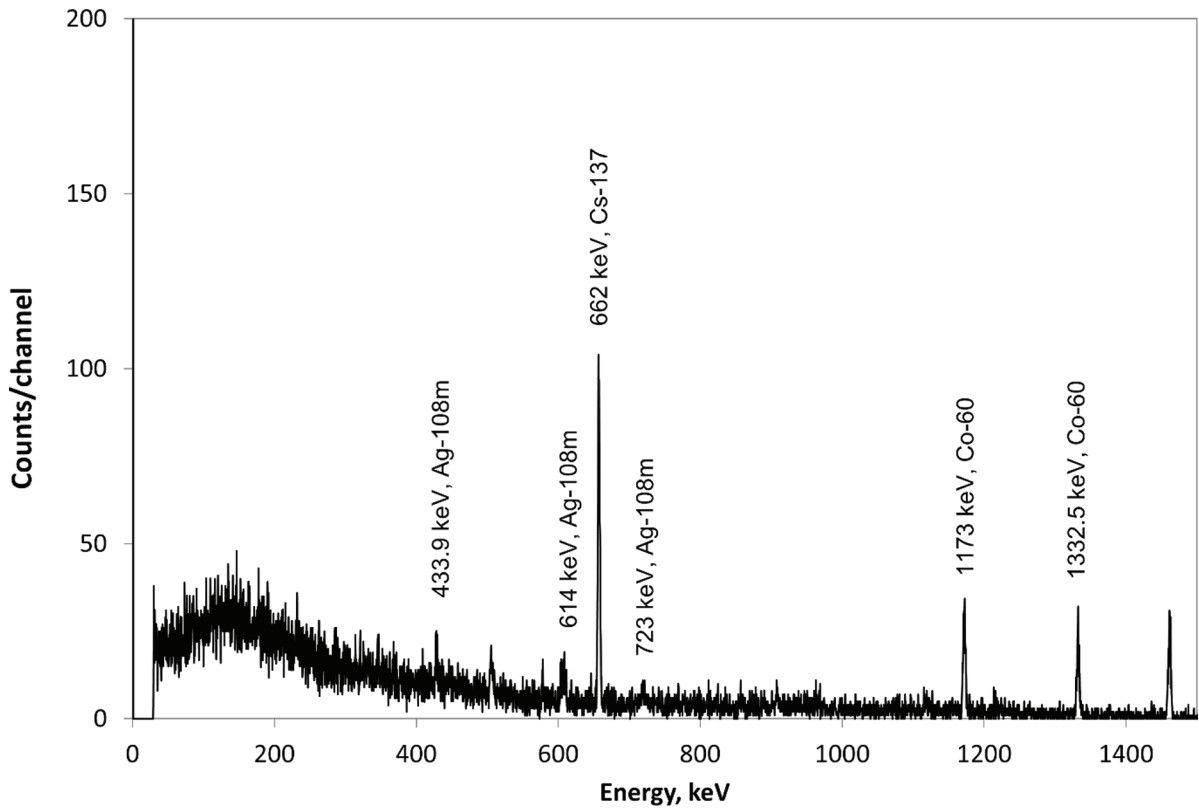

**Fig 6. Gamma spectrum of the sample taken from the beryllium chloride storage.**

above the permissible activity level, starting from which the radioactive material use may be unlimited. The results of measurements of the activity of the samples taken from different installation units confirmed the fact of a chemical transport reaction implementation, during which purified beryllium chloride is isolated from the composition of the chlorination products of the irradiated beryllium, the cobalt chloride absorbs on the appropriate filter, and a part of the tritium chloride dissolves in the tritium chloride storage tank. Based on the results obtained in the process of $\beta$-spectrometry of the water samples taken from the tritium chloride storage tank, it has been shown that the adopted tritium chloride capture scheme is efficient, but the tritium chloride absorption efficiency does not exceed 3% of the total amount of tritium chloride that could be formed during the irradiated beryllium purification. It can be assumed that the efficiency of trapping tritium chloride in the storage can be increased by increasing the time of gaseous tritium chloride interaction with water, which can be achieved by multiple passing of the gas through water using bubbling devices, as well as by using multi-stage tritium chloride traps. When determining the temperature conditions of the technological process of irradiated beryllium purification, it should be taken into consideration that the chemical reactions occurring in the installation are exothermic in nature, and the thermal energy release should be accounted for in the industrial installation design. The interaction of chlorine with iron, leading to ferric chloride formation, is a negative factor that reduces the installation reliability and operability, and ultimately determines the need to search for alternative structural materials and design solutions. The zone melting method can be used as an additional method for further decontamination of beryllium chloride from

**Table 1. Results of determining the specific activity of the samples.**

| Place of Sampling | Beryllium Mass (g) | Specific Activity (Bq/g(Be)) | | |
|---|---|---|---|---|
| | | $^{60}$Co | $^{108m}$Ag | $^{137}$Cs |
| Initial sample | 70.3 | $(3.8 \pm 0.2) \cdot 10^4$ | $(4.3 \pm 0.3) \cdot 10^3$ | $(6.1 \pm 0.4) \cdot 10^3$ |
| Reaction chamber | 0.03 | $400 \pm 24$ | $200 \pm 12$ | $(2.4 \pm 0.1) \cdot 10^3$ |
| Beryllium chloride storage | 0.03 | $70 \pm 4.9$ | $15 \pm 1.1$ | $200 \pm 14$ |

**Table 2. Results of beta-spectrometric measurements of the solution samples from the tritium chloride storage tank.**

| No. | Place of Sampling | Specific Activity of $^3$H (Bq/kg) |
|---|---|---|
| 1 | The standard (dist. water) | < 8 |
| 2–4 | Tritium chloride storage tank | $(22.35 \pm 3.1) \cdot 10^6$ |
| | | $(21.04 \pm 2.7) \cdot 10^6$ |
| | | $(21.04 \pm 2.5) \cdot 10^6$ |

impurities [25,26], as it is applicable to chloride salts in general. The revivification of beryllium chloride into metallic beryllium can be carried out by well-known industrial methods, such as the electrolytic method and the method of high-temperature decomposition.

## Conclusion

The study of the process of the irradiated beryllium reflector decontamination in a chlorine environment made it possible to determine the installation structure and technical parameters intended for the implementation of the technological process of beryllium purification from radioactive impurities and to establish the operating modes of the installation's main units. This method is based on the application of transport reactions using beryllium chlorination, followed by the release of beryllium chloride from the composition of the chlorides of radioactive impurities. The measurements were performed using gamma-spectrometric and $\beta$-spectrometric methods. During the measurements, the radionuclides $^{60}$Co, $^{108m}$Ag, $^{137}$Cs and $^3$H, were identified and quantitatively characterized. The experimental results of measuring the purification coefficients of irradiated beryllium from radionuclides using chloride technology have confirmed the possibility of reducing beryllium radioactivity by orders of magnitude. However, the beryllium purified in this way still has a specific activity above the limit of permissible values for its unrestricted use. Nevertheless, such a metal can certainly be reused for nuclear power facilities.

## Author contributions

**Conceptualization:** Yuliya Baklanova.

**Investigation:** Yuliya Baklanova.

**Writing – original draft:** Yuliya Baklanova.

**Writing – review & editing:** Yuliya Baklanova.

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
