## [Decision Letter · Decision Letter 0]

28 Jan 2025

PONE-D-25-00607Studying the decontamination process of an irradiated beryllium reflector in a chlorine environmentPLOS ONE

Dear Dr. Baklanova,

Thank you for submitting your manuscript to PLOS ONE. After careful consideration, we feel that it has merit but does not fully meet PLOS ONE’s publication criteria as it currently stands. Therefore, we invite you to submit a revised version of the manuscript that addresses the points raised during the review process.

We look forward to receiving your revised manuscript.

Kind regards,

Mohammad Alrwashdeh

Academic Editor

PLOS ONE

Journal requirements: When submitting your revision, we need you to address these additional requirements. 1. Please ensure that your manuscript meets PLOS ONE's style requirements, including those for file naming. The PLOS ONE style templates can be found at https://journals.plos.org/plosone/s/file?id=wjVg/PLOSOne_formatting_sample_main_body.pdf and https://journals.plos.org/plosone/s/file?id=ba62/PLOSOne_formatting_sample_title_authors_affiliations.pdf. 2. Please update your submission to use the PLOS LaTeX template. The template and more information on our requirements for LaTeX submissions can be found at http://journals.plos.org/plosone/s/latex. 3. Thank you for stating the following financial disclosure:  [The work was financed by the Science Committee of the Ministry of Science and Higher Education of the Republic of Kazakhstan within the framework of IRN BR21882185 “Research to support the creation and safe operation of a nuclear power plant in the Republic of Kazakhstan”].  Please state what role the funders took in the study.  If the funders had no role, please state: ""The funders had no role in study design, data collection and analysis, decision to publish, or preparation of the manuscript."" If this statement is not correct you must amend it as needed. Please include this amended Role of Funder statement in your cover letter; we will change the online submission form on your behalf. 4. We note that your Data Availability Statement is currently as follows: [All relevant data are within the manuscript and its Supporting Information files.] Please confirm at this time whether or not your submission contains all raw data required to replicate the results of your study. Authors must share the “minimal data set” for their submission. PLOS defines the minimal data set to consist of the data required to replicate all study findings reported in the article, as well as related metadata and methods (https://journals.plos.org/plosone/s/data-availability#loc-minimal-data-set-definition). For example, authors should submit the following data: - The values behind the means, standard deviations and other measures reported;- The values used to build graphs;- The points extracted from images for analysis. Authors do not need to submit their entire data set if only a portion of the data was used in the reported study. If your submission does not contain these data, please either upload them as Supporting Information files or deposit them to a stable, public repository and provide us with the relevant URLs, DOIs, or accession numbers. For a list of recommended repositories, please see https://journals.plos.org/plosone/s/recommended-repositories. If there are ethical or legal restrictions on sharing a de-identified data set, please explain them in detail (e.g., data contain potentially sensitive information, data are owned by a third-party organization, etc.) and who has imposed them (e.g., an ethics committee). Please also provide contact information for a data access committee, ethics committee, or other institutional body to which data requests may be sent. If data are owned by a third party, please indicate how others may request data access. 5. Please remove your figures from within your manuscript file, leaving only the individual TIFF/EPS image files, uploaded separately. These will be automatically included in the reviewers’ PDF.

Reviewers' comments:

Reviewer's Responses to Questions

**Comments to the Author**

1. Is the manuscript technically sound, and do the data support the conclusions?

Reviewer #1: Yes

Reviewer #2: Yes

2. Has the statistical analysis been performed appropriately and rigorously? 

Reviewer #1: Yes

Reviewer #2: Yes

3. Have the authors made all data underlying the findings in their manuscript fully available?

Reviewer #1: Yes

Reviewer #2: Yes

4. Is the manuscript presented in an intelligible fashion and written in standard English?

Reviewer #1: Yes

Reviewer #2: Yes

5. Review Comments to the Author

Reviewer #1: I have reviewed the article titled "Studying the decontamination process of an irradiated beryllium reflector in a chlorine environment". The article present an interesting study to decontaminate the beryllium used in nuclear reactor. My comments are as follows

1. There are few typos in the article. For examples in abstract it is written "H3 including tritium" when it should be "including H3 tritium". For the gamma radiation spectrum graphs, the ordinate and abscissa titles are not in English. Page 2, 118mAg is wrong. Similarly page 4, 60CoCl2 is not written correctly in 3rd paragraph.

2. In figure 1, 614 and 723 keV peaks are labelled as Ag-110 but in all the other figures the peaks are labelled as Ag-108.

3. In results and discussion, it has been mentioned that tritium chloride capture efficiency is only 3%. additional studies are needed to verify the "assumption" that this efficiency can be increased with contact time of gaseous tritium chloride with water and using multi-stage tritium chloride traps.

4. Page 5, iron chloride transportation in the system, which kind of candidate materials you have in mind to avoid this problem.

5. First line of introduction, "the beryllium loses it unique properties as a neutron moderator and reflector". I think this needs to be modified or relevant reference needs to be provided which shows that there is decreases in beryllium's moderation and reflection capabilities after irradiation.

Reviewer #2: The manuscript focuses on an important topic in nuclear waste management - the decontamination of irradiated beryllium reflectors using chlorination processes. The underlying methodology is systematic and based on established chemical principles, validated through experimental measurements. The authors provide detailed documentation of the developed chlorination process and the results obtained from their pilot installation.

The implementation of this decontamination approach addresses a significant challenge in nuclear waste management, particularly given the increasing global inventory of irradiated beryllium. Taking this into consideration, I recommend that the manuscript be considered for publication in PLOS ONE, subject to the following revisions:

1. A major limitation of this study concerns the experimental validation which is based on only one type of sample from the JMTR reflector. It would be important to justify why this sample is considered representative and how the results could be extrapolated to other types of irradiated beryllium reflectors. A discussion of potential variations expected with different types of samples would significantly strengthen the scope of the study.

2. The chosen operational parameters, particularly the temperature of 730°C and the 30 chlorination cycles, require more thorough justification. The authors should explain how these parameters were optimized and present the relative efficiency of each cycle. This information is crucial for understanding the robustness of the process and its potential for optimization.

3. The formation of iron chloride due to corrosion represents a significant technical challenge that is not sufficiently addressed. The authors should discuss in more detail possible alternatives in terms of construction materials, evaluate the impact of this corrosion on the installation's lifespan, and propose solutions to minimize this problem, particularly in the perspective of larger-scale application.

4. Regarding formal aspects, the manuscript requires several improvements: sections should be clearly numbered, figures lack detailed captions and some are of insufficient quality, particularly the gamma spectra. Error bars should be added to all graphs and results tables. The bibliography should be enriched with more recent references on beryllium decontamination methods.

These clarifications and additions would make the manuscript more complete and relevant to the scientific and industrial community.

6. PLOS authors have the option to publish the peer review history of their article (what does this mean?). If published, this will include your full peer review and any attached files.

Reviewer #1: No

Reviewer #2: No

---

## [Author Response · Author response to Decision Letter 1]

21 Mar 2025

Response to Reviewer 1

General Comment:

"I have reviewed the article titled 'Studying the decontamination process of an irradiated beryllium reflector in a chlorine environment'. The article presents an interesting study to decontaminate the beryllium used in nuclear reactors. My comments are as follows."

Author Response:

I sincerely appreciate your thoughtful review and constructive feedback. Your insights have helped improve the quality of the manuscript. I am grateful for your comments and have carefully addressed each of your concerns in detail below.

Remark No. 1:

"There are a few typos in the article. For example, in the abstract, it is written 'H3 including tritium' when it should be 'including H3 tritium'.

For the gamma radiation spectrum graphs, the ordinate and abscissa titles are not in English.

Page 2, 118mAg is wrong.

Similarly, on page 4, 60CoCl2 is not written correctly in the third paragraph."

Author Response to Remark No. 1:

I apologize for the inaccuracies. I have corrected all typos, including the phrase in the abstract, and thoroughly reviewed the text of the manuscript.

The axis titles of all gamma radiation spectrum graphs have been translated into English for consistency.

The incorrect isotope notation on page 2 has been corrected to Ag-108m.

The chemical formula for cobalt chloride on page 4 has also been corrected to 60CoCl2.

Remark No. 2:

"In Figure 1, the 614 and 723 keV peaks are labeled as Ag-110, but in all the other figures, the peaks are labeled as Ag-108."

Author Response to Remark No. 2:

As you correctly pointed out, the correct isotope should be Ag-108m. I have now made the necessary corrections in Figure 1 to ensure consistency across all figures.

Remark No. 3:

"In the results and discussion, it has been mentioned that tritium chloride capture efficiency is only 3%. Additional studies are needed to verify the 'assumption' that this efficiency can be increased with the contact time of gaseous tritium chloride with water and using multi-stage tritium chloride traps."

Author Response to Remark No. 3:

I appreciate the reviewer’s attention to this point. In the manuscript, I have stated that tritium chloride absorption efficiency does not exceed 3% and suggested that efficiency could be improved by increasing the contact time of gaseous tritium chloride with water, employing bubbling devices, or using multi-stage tritium chloride traps.

The relevant calculations have been provided in the following reference:

Yu. Baklanova, A. Vurim, V. Kotov, A. Sitnikov, L. Chernova. Features of Chloride Technology for Processing Irradiated Beryllium. Polzunovsky Bulletin No. 3, 2019. [https://doi.org/10.25712/ASTU.2072-8921.2019.03.016] (In Russian).

The current tritium capture system was primarily designed for proof-of-concept demonstration, and future research will focus on optimizing the system parameters to enhance tritium retention. Additionally, my future research, titled "Study of Corrosion Behavior of Austenitic Steel 12X18N10T in a Chlorine Medium," which has been submitted to the Journal of Corrosion and Materials Degradation, further explores this topic. I hope that the findings from this ongoing research will provide a more comprehensive understanding of the corrosion mechanisms involved.

Remark No. 4:

"Page 5, iron chloride transportation in the system, which kind of candidate materials you have in mind to avoid this problem"

Author Response to Remark No. 4

I appreciate the reviewer’s question regarding potential materials to mitigate iron chloride formation. As noted in the manuscript, the formation of iron chloride results from the interaction of chlorine with stainless steel components, leading to corrosion.

To address this issue, quartz glass, borosilicate glass, and nickel-based alloys are promising candidates for improving resistance to chlorine-induced corrosion. However, I acknowledge that this assumption requires further experimental verification to determine the long-term stability of these materials under operational conditions.

Remark No. 5:

"First line of introduction, "the beryllium loses it unique properties as a neutron moderator and reflector". I think this needs to be modified or relevant reference needs to be provided which shows that there is decreases in beryllium's moderation and reflection capabilities after irradiation."

Author Response to Remark No. 5:

I have revised the introduction to provide a more precise explanation of the degradation of beryllium's neutron-moderating and reflecting properties under prolonged irradiation. Specifically, I have included references to studies that demonstrate how neutron-induced swelling, helium and tritium accumulation, and impurity activation affect beryllium’s structural and neutron transport properties, ultimately impacting its performance in reactors.

Response to Reviewer 2

General Comment:

*"The manuscript focuses on an important topic in nuclear waste management—the decontamination of irradiated beryllium reflectors using chlorination processes. The underlying methodology is systematic and based on established chemical principles, validated through experimental measurements. The authors provide detailed documentation of the developed chlorination process and the results obtained from their pilot installation.

The implementation of this decontamination approach addresses a significant challenge in nuclear waste management, particularly given the increasing global inventory of irradiated beryllium. Taking this into consideration, I recommend that the manuscript be considered for publication in PLOS ONE, subject to the following revisions."*

Author Response:

Dear Reviewer,

I sincerely appreciate your careful review of the manuscript and your insightful comments. I am grateful for these detailed recommendations, which have allowed me to improve the clarity and scientific depth of the work. I hope the revisions sufficiently address your concerns and appreciate your support in improving the manuscript.

Below, I provide my responses to each of your points.

Comment 1:

"A major limitation of this study concerns the experimental validation, which is based on only one type of sample from the JMTR reflector. It would be important to justify why this sample is considered representative and how the results could be extrapolated to other types of irradiated beryllium reflectors. A discussion of potential variations expected with different types of samples would significantly strengthen the scope of the study."

Author Response:

I acknowledge the reviewer’s concern regarding the representativeness of the JMTR reflector sample used in this study. The choice of this sample was based on its well-documented, certified irradiation history, operational conditions, and composition, making it a suitable case study for evaluating the chlorination-based decontamination process, as outlined in Ref. 18.

To address the extrapolation aspect, I have expanded the discussion in the revised manuscript to include potential variations expected when applying this process to other irradiated beryllium reflectors, considering factors such as neutron fluence, impurity accumulation, and microstructural changes that may influence the chlorination efficiency.

Comment 2:

"The chosen operational parameters, particularly the temperature of 730°C and the 30 chlorination cycles, require more thorough justification. The authors should explain how these parameters were optimized and present the relative efficiency of each cycle. This information is crucial for understanding the robustness of the process and its potential for optimization."

Author Response:

I appreciate this suggestion and have revised the manuscript to better explain the selection of 730°C as the reaction temperature and the 30-cycle process. These parameters were determined based on prior computational and experimental data (see Ref. 17 and 24) and thermodynamic analysis, ensuring efficient chlorination while preventing excessive volatilization losses.

Additionally, I have expanded the discussion on the efficiency of each cycle by summarizing the results of preliminary optimization studies, which demonstrated that 730°C provides a balance between decontamination effectiveness and minimization of unwanted side reactions.

The following sentences inserted into the text of the manuscript.

The installation's performance is calculated based on the fact that the operation of the proposed device consists of two phases: active and passive. After heating each unit of the installation to operating temperatures is supplied into the reaction chamber during the active phase of operation. The hot beryllium chloride flow reaches the filter, gradually increasing the temperature of its working elements. The active phase ends when the temperature of the filter's working elements reaches 700°C.

During the passive phase, chlorine supply is stopped, and the direct-flow system is cooled with argon,

The selection of operating temperatures is based on the phase transition temperatures of the obtained chlorides [22, 25]: the boiling point of beryllium chloride is 500°C, and the melting point of cobalt chloride is 724°C. The duration of the active phase was determined by the time required to heat nickel rods by 200°C in a flow of hot beryllium chloride, which was 6 minutes, while the duration of the passive phase is 5 minutes.

Comment 3:

"The formation of iron chloride due to corrosion represents a significant technical challenge that is not sufficiently addressed. The authors should discuss in more detail possible alternatives in terms of construction materials, evaluate the impact of this corrosion on the installation's lifespan, and propose solutions to minimize this problem, particularly in the perspective of larger-scale application."

Author Response:

The iron chloride formation due to corrosion poses a significant challenge in maintaining the integrity of the installation. In response to the reviewer’s request for more details:

Alternative Materials: Quartz glass, borosilicate glass, and nickel-based alloys are considered promising candidates for improved corrosion resistance. However, further experimental validation is required to assess their long-term stability in chlorination environments.

Impact on Installation Lifespan: The formation of iron chloride can lead to material degradation, reducing the operational lifespan of reactor components.

Solutions for Large-Scale Application: Design modifications such as improved gas flow control, protective coatings, and alternative construction materials may help mitigate this issue.

These points have been inserted into the revised manuscript to provide a more comprehensive discussion of long-term system stability.

Comment 4:

"Regarding formal aspects, the manuscript requires several improvements: sections should be clearly numbered, figures lack detailed captions and some are of insufficient quality, particularly the gamma spectra. Error bars should be added to all graphs and results tables. The bibliography should be enriched with more recent references on beryllium decontamination methods.

These clarifications and additions would make the manuscript more complete and relevant to the scientific and industrial community."

Author Response:

I appreciate these recommendations and have implemented the following revisions:

Sections have been numbered for better organization.

Figure captions have been expanded to provide clearer descriptions of the data presented.

Gamma spectra figures have been improved, and higher-quality versions have been included.

Error bars have been added to tables to better represent measurement uncertainties.

The references section has been updated with additional recent sources on beryllium decontamination methods to enhance the manuscript’s relevance to current research.

---

## [Editor Report · Decision Letter 1]

26 Mar 2025

Studying the decontamination process of an irradiated beryllium reflector in a chlorine environment

PONE-D-25-00607R1

Dear Dr. Baklanova,

We’re pleased to inform you that your manuscript has been judged scientifically suitable for publication and will be formally accepted for publication once it meets all outstanding technical requirements.

Kind regards,

Mohammad Alrwashdeh

Academic Editor

PLOS ONE
---

## [Editor Report · Acceptance letter]

PONE-D-25-00607R1

PLOS ONE

Dear Dr. Baklanova,

I'm pleased to inform you that your manuscript has been deemed suitable for publication in PLOS ONE. Congratulations! Your manuscript is now being handed over to our production team.

Kind regards,

on behalf of

Dr. Mohammad Alrwashdeh

Academic Editor

PLOS ONE